# Development of a New Aggregation Method to Remove Nanoplastics from the Ocean: Proof of Concept Using Mussel Exposure Tests

**DOI:** 10.3390/biomimetics9050303

**Published:** 2024-05-18

**Authors:** Antonio Cid-Samamed, Catarina S. E. Nunes, Cristina Lomas Martínez, Mário S. Diniz

**Affiliations:** 1Physical Chemistry Department, Faculty of Sciences, University of Vigo, Campus de As Lagoas S/N, 32004 Ourense, Spain; 2i4HB—Associate Laboratory Institute for Health and Bioeconomy, NOVA School of Science and Technology, NOVA University Lisbon, 2819-516 Caparica, Portugalmesd@fct.unl.pt (M.S.D.); 3UCIBIO, Chemistry Department, NOVA School of Science & Technology, Universidade NOVA de Lisboa, 2819-516 Caparica, Portugal; 4Department of Biotechnology and Environmental Protection, Estación Experimental del Zaidín, Consejo Superior de Investigaciones Científicas, 18008 Granada, Spain; cristina.lomas@eez.csic.es

**Keywords:** microplastics, nanoplastics, polystyrene, ionic liquid, *Mytilus galloprovincialis*, oxidative stress

## Abstract

The overproduction and mismanagement of plastics has led to the accumulation of these materials in the environment, particularly in the marine ecosystem. Once in the environment, plastics break down and can acquire microscopic or even nanoscopic sizes. Given their sizes, microplastics (MPs) and nanoplastics (NPs) are hard to detect and remove from the aquatic environment, eventually interacting with marine organisms. This research mainly aimed to achieve the aggregation of micro- and nanoplastics (MNPs) to ease their removal from the marine environment. To this end, the size and stability of polystyrene (PS) MNPs were measured in synthetic seawater with the different components of the technology (ionic liquid and chitosan). The MPs were purchased in their plain form, while the NPs displayed amines on their surface (PS NP-NH_2_). The results showed that this technology promoted a significant aggregation of the PS NP-NH_2_, whereas, for the PS MPs, no conclusive results were found, indicating that the surface charge plays an essential role in the MNP aggregation process. Moreover, to investigate the toxicological potential of MNPs, a mussel species (*M. galloprovincialis*) was exposed to different concentrations of MPs and NPs, separately, with and without the technology. In this context, mussels were sampled after 7, 14, and 21 days of exposure, and the gills and digestive glands were collected for analysis of oxidative stress biomarkers and histological observations. In general, the results indicate that MNPs trigger the production of reactive oxygen species (ROS) in mussels and induce oxidative stress, making gills the most affected organ. Yet, when the technology was applied in moderate concentrations, NPs showed adverse effects in mussels. The histological analysis showed no evidence of MNPs in the gill’s tissues.

## 1. Introduction

Arising from the lifestyle of modern society, plastics for single use have been gaining particular interest [1]. Many disposable products have polystyrene (PS) as a base material [2]. Properties like its thermostability, hardness, low water absorption, and absence of odor and taste make this homopolymer advantageous for the disposable plastics industry, especially for food storage products, packaging materials, and household goods applications [2].

On the other hand, PS presents sensitivity to UV radiation, which may affect some of its mechanical and chemical properties, and therefore, PS is rarely used in its singular form [2,3]. Hence, additives such as plasticizers and UV stabilizers are often applied during or after the PS synthesis process to reinforce the material quality and extend its life span. Hindered amine light stabilizers (HALS) are one of the most used UV stabilizers in plastics since it has been revealed that this type of amine group confers long-term photo-oxidation stabilization and a higher weather resistance [2,4]. Nowadays, PS is considered one of the leading traded plastics [5,6], and despite generally being manufactured for disposal, PS products have high durability and resistance, thus having a slow degradation and consequently accumulating in the environment [5].

Every year, approximately 8 million tons of plastic enter the marine environment due to mismanagement or careless disposal of this waste [3,7]. Once in the environment, plastics become subjected to photophysical, mechanical, and biological factors such as UV radiation, wind, and microorganisms, leading to fragmentation [1,3]. As a result of this process, plastics that initially presented macroscopic dimensions may give rise to microscopic or even nanoscopic debris, termed microplastics (MPs, <5 mm) and nanoplastics (NPs, <100 nm), respectively [1]. Besides this plastic’s debris originating from fragmentation and degradation processes, which are called secondary microplastics, it is also worth mentioning the primary microplastics. Primary MPs are plastics intentionally manufactured to present a microscopic size [2].

Thomson et al. [8] introduced the term microplastic by providing the first experimental data from sediment samples around the United Kingdom. Moreover, these authors proved an increase in the abundance of MPs over thirty years. Due to their tiny dimensions and light weight, micro- and nanoplastics (MNPs) are easily transported across terrestrial subsystems, making them very persistent and mobile chemicals. In terms of the marine environment, plastic debris distribution is continuously triggered by ocean currents and gyres. The most affected areas are the seacoasts near urban and industrialized territories and gyre centers. In 1972, Theodore Merrel found plastic litter on Amchitka Island in Alaska [9], and more recently, various scientists reported the presence of MNPs in the Arctic [10,11,12], indicating that these anthropogenic pollutants have already reached remote regions. All these pieces of evidence reveal that plastic debris has been accumulating even in the most pristine locations, thus raising critical awareness.

Ocean water is a complex matrix of different components such as salts, minerals, metals, organic and inorganic matter, and a few atmospheric gases dispersed as a liquid colloid [13]. An aqueous colloid is a mixture of small materials or substances dispersed uniformly through other solutes in a fluid. There are various types of colloidal systems according to the state of matter; however, when considering plastic debris with micro- and nanoscopic dimensions as a solute, the respective colloidal solution is a sol, which is a mixture of tiny (<1 µm) solid materials in an aqueous medium [14]. This type of colloid presents a continuous phase, the liquid medium of suspension, and a dispersed phase, the suspended colloidal particles. Moreover, a colloid is considered stable when particles in a solution are randomly well dispersed and unstable when particles collide, leading to aggregates forming. This aggregation process of colloidal particles is commonly referred to as coagulation. Furthermore, the dispersed colloidal particles have a spontaneous thermal motion known as Brownian movement, generally described as the random motion of particles arising from collisions with surrounding molecules [13]. This thermodynamic aspect of colloidal systems triggers a constant particle diffusion.

According to the Derjaguin−Landau−Verwey−Overbeek (*DLVO*) theory [15,16], the coagulation process responsible for colloidal particles’ aggregation is controlled by van der Waals attractive forces and electrostatic/Coulombic repulsive forces, which are respectively dipole–dipole interactions and electrical double layer interactions [17]. Thus, the balance of these opposite forces will determine the thermodynamic state of the colloidal system, directly associated with Brownian motion and, ultimately, the transition between its aggregative and dispersive phase [3]. Despite being related to the thermodynamic equilibrium, it is worth mentioning that colloidal stability is obtained not when the colloid is thermodynamically stable, but rather when its coagulation rate is nearly zero [15]. Hence, for particles to aggregate upon collision, the resultant kinetic energy has to overcome this maximum threshold value [13,17,18,19]. Once this phenomenon happens, the interaction energy drops steeply, reaching the first coagulation minimum (primary minimum). In this state, colloidal particles develop an irreversible cluster structure, and the colloid is considered unstable. Accordingly, it can establish a critical coagulation concentration (CCC), defined as the minimum electrolyte concentration necessary for the total destabilization of a colloid [3]. This value directly influences the coagulation rate and the energy barrier height [20,21]. It is indeed considered a chief parameter to tailor the potential energy of a colloid [20]. Furthermore, as described by the Schulze−Hardy rule, the greater the valence of the counterions, the ions with the opposite charge to the colloidal particles, the lower the CCC [22,23]. Therefore, if the ionic strength is above the CCC value and the counterions in the solution are multivalent, overall, the coagulation rate and particle–particle attachment efficiency increase.

Besides these colloidal aspects, particle behavior in natural colloids, such as seawater, is also intimately related to particle–particle properties, namely their size, density, chemical composition, and surface charge [24,25]. Among these characteristics, surface charge, in particular, plays a significant role in the affinity between particles and ions in solution [18,26]. Particles can have intrinsic charges on their surface and be developed by exposing the ions from their constituent elements via surface degradation mechanisms. Moreover, under specific and favorable conditions, particle surface charges may even be acquired by ionic adsorption [25].

Beyond those colloidal and particle considerations, there is another issue of great importance regarding seawater as a medium: the presence of natural organic matter (NOM). Composed of plants or animal remains, egesta, microorganisms, and extracellular polymeric substances, NOM has a high protein content with active functional groups that enable interactions with other suspended particles [1,27]. Interestingly, such biological substances can stabilize micro- and nanoplastics by coating their surface and forming an outer layer [1]. These particle features and medium conditions will influence the MPs’ and NPs’ physicochemical transformations in seawater, subsequently determining the final fate of these particles in the ecosystem. 

Microplastics are a significant concern for the marine ecosystem since such dimensions favor their internalization by a wide range of animals. It has been reported that microplastics have been detected in the gastrointestinal tract of several aquatic species [28,29,30,31,32]. Although MNPs are not digested or degraded by these organisms, since they lack enzymes capable of processing synthetic polymers, the chemicals from the plastics or contaminants present in the environment that attach to MPs can leach out during their retention time within the organisms [32,33].

Combined with microplastic breakdown triggered by environmental factors, there is a build-up process in which other marine pollutants are absorbed on MNPs’ surface, which can further increase their potential toxicological hazard. Persistent organic pollutants (POPs) include a broad range of toxic substances encountered in seawater that adhere to the plastic surface [32,34]. These contaminants’ attachment process is enhanced by the increase in MNPs’ specific surface area, the exposition of the polymer backbone compounds, which may increase MNPs’ reactivity, or simply because of the intrinsic hydrophobic nature of both polymers and pollutants [1,35]. These contaminants, if released inside the organisms, can cause damage to the immune system [36]. Furthermore, some ingested MNPs can accumulate in various body structures without degradation. Moreover, Browne et al. reported for the first time the translocation of MPs from the gut to the circulatory system [37]. As previously mentioned, NOM in the seawater can attach to the MNP surface and become wrapped in these substances. Due to this new organic coating, MNPs become optimal substrates for the development of microbes/microorganisms (e.g., bacteria, fungi, and algae) [34]. Consequently, these “biofouled” MNPs can be misidentified as food, which induces ingestion. For instance, filter-feeding animals regularly ingest these particles, firstly due to the large volumes of water they filter, but also because they are not able to scrutinize the food from MNPs [1,32,38]. Moreover, this propensity of NOM to attach to the surface of MNPs will create changes in MNPs’ surface chemistry, ultimately influencing their toxicity [27]. Additionally, the stabilizing effect of NOM makes MNPs more bioavailable and, therefore, even more prone to internalization by organisms [1]. To some extent, the toxicity of this plastic debris in the marine ecosystem can create an imbalance in the biogeochemical cycles and compromise biodiversity [32,34].

Furthermore, many marine organisms in constant contact with MNPs, like seafood and fish, are present in our diet; therefore, MNPs can represent a significant threat to food safety and, consequently, a potential health hazard for humans. These microscopic particles have already been detected in human lungs and feces [12].

Considering the relevance that MNPs have been acquiring in marine litter science, methods have been developed to capture and analyze these contaminants. The Neuston net is the standard equipment used in situ that provides valuable samples of MPs in the surface water [39,40]. Although the Neuston net presents some limitations, considering that the mesh size determines the size and shape of particles captured, this method can only remove MPs larger than 300 µm [41,42]. 

Filtration techniques have been developed and widely applied [40]. For instance, Lenz and Labrenz created a filtration system that allows the sampling of particles up to 10 µm from the surface and water column [43]. However, these filtration methods tend to be time-consuming and usually limit the sampling of small MNPs [39]. Additionally, these sampling strategies generally imply additional treatments like density separation or wet oxidation [40]. More precise techniques such as crossflow ultrafiltration and asymmetrical flow field-flow fractionation (AF4) enable the capture of nano-sized particles [42,44,45].

Recently, Rhein et al. developed a magnetic-seeded filtration based on the aggregation of magnetic seed particles to MPs, which subsequently were collected through a separation matrix consisting of fine ferromagnetic wires [46]. Furthermore, bioremediation processes involving microorganisms capable of degrading plastics have been extensively studied. Indeed, several authors have reported different bacterial and fungal species capable of decomposing different types of MPs [47]. Lipases, esterases, and proteases are some of the main groups of enzymes involved in these degradative mechanisms [48,49].

The goal of this research was to propose and test a new method (or technology) to use a biocompatible surfactant and polyelectrolyte to ease—by mimicking the behavior of natural organic matter when adsorbing onto MNPs’ surfaces—the aggregation of NPs into MPs and MPs into mesoplastics to help remove them from aquatic environments, or at least achieve enough aggregation to diminish their bioavailability, which will decrease their adverse effects on aquatic biota.

## 2. Materials and Methods

### 2.1. Ionic Liquids as Surface-Active Agents and Chitosan as Polyelectrolyte

Surface-active agents, usually referred to as surfactants, are chemicals comprised of two parts, a hydrophilic head group and a hydrophobic tail, and can be classified according to their ionic or non-ionic properties [50]. These amphiphilic molecules have unique physical and chemical properties, particularly their self-assembly in aqueous solutions [51]. When surfactant molecules are added to water above a specific concentration known as the critical micellar concentration (CMC), they tend to aggregate into micelle-like structures [51]. This molecule rearrangement limits the direct contact of the surfactant’s hydrophobic tail with water, thus making micelles thermodynamically more stable [50].

Ionic liquids (ILs) have been reported as a new class of surfactants [52]. ILs are molten salts below 100 °C, entirely made of ions [50], and usually composed of organic cations combined with organic or inorganic anions [53]. Owing to their surface-active properties, ILs with long-enough alkyl chains (hydrophobic tail) can encapsulate hydrophobic compounds inside their micelle core, enabling their extraction from aqueous solutions [54]. The proposed aggregation method uses a surfactant (FIL) and polyelectrolyte (chitosan) to (bio-)mimic the behavior of NOM adsorbed onto MNPs’ surfaces under actual environmental conditions.

### 2.2. Biological Model: Mediterranean Mytilus galloprovincialis

Bioindicators are defined as species capable of biomonitoring the quality status of an environment in terms of evaluating contaminants’ content, bioavailability, and toxicity [55]. For instance, mussels are one of the chief groups of organisms used for the assessment of many pollutants and toxic substances in the marine environment, including MNPs [56,57,58]. Due to their non-selective filter-feeding strategy, mussels internalize large volumes of water, which are frequently in contact with those pollutant particles [1,32]. Furthermore, on the food web, they are generally primary consumers, and therefore, they continuously transfer matter throughout higher trophic levels, which may lead to biomagnification [59]. *Mytilus galloprovincialis*, commonly known as Mediterranean mussel, is a commercially relevant bivalve mollusk that inhabits the intertidal and subtidal areas of rocky shores. Despite being native to the Mediterranean and Eastern Atlantic, nowadays, this mussel species has a wide range of distribution and can be found in South America, South Africa, Japan, and Australia [60,61]. Its wide distribution, feeding behavior, and commercial value make these organisms good biological models for MNP hazard and toxicological assessments [62]. Indeed, various authors have used *M. galloprovincialis* for MNP studies and revealed that these particles could induce a stress response [32,56].

### 2.3. MNPs and Aggregation Method Characterization

#### 2.3.1. Stock Solutions Preparation

The characterization of MPs and NPs was carried out both in ultrapure water (Milli-Q^®^) and in synthetic seawater, with both solvents filtered using a 0.20 µm filter (PTFE syringe filter 0.20 µm, Millex^®^-LG Millipore Ibérica, Madrid, Spain). Commercial amine-modified polystyrene latex nanospheres (NPs) and polystyrene microspheres (MPs) with a nominal diameter of 50 nm and 1 µm, respectively, were manufactured by Sigma-Aldrich (Sant Louis, MO, USA). MPs were purchased in an aqueous solution of 50 mg/mL and diluted to obtain a stock solution of 0.2 mg/mL in both solvents. NPs were acquired in an aqueous solution of 250 mg/mL and diluted to obtain a stock solution of 0.1 mg/mL in each solvent. Both stock solutions were then ultrasonicated (10 min, 35 kHz at room temperature) using an ultrasonic bath (J-P Selecta Ultrasounds HD, Barcelona, Spain) and stored in the dark at 4 °C until further use. Chitosan of medium molecular weight (300 g/mol) from Sigma Aldrich (C.A.S. Number: 9012-76-4, Germany) was prepared in Milli-Q water to a final concentration of 0.3 g/mL, filtered with a 0.20 µm filter, and mixed with the aid of a vortex (I@LAB MX-S). Fluorinated ionic liquid (FIL) (ethyl-3-methylpyridinium perfluorobutanesulfonate, [EtMepy]-[(PFBu)SO_3_]) from Ana B. Pereiro’s research group was acquired with a concentration of 1453.6 mg/mL (>99% mass fraction purity) [63].

#### 2.3.2. Dynamic Light Scattering (DLS) and ζ-Potential

Particle size distribution, average hydrodynamic diameter (*D_h_*, Z-average and polydispersity index, PDI), and colloidal stability (ζ-potential and electrophoretic mobility, µ) measurements were determined by dynamic light scattering (DLS), and ζ-potential with a Zetasizer Nano Series ZEN3600 (Malvern, U.K.) with a 633 nm laser, at a scattering angle of 90°. 

Dynamic light scattering (DLS) is a non-invasive technique based on measuring the Brownian motion of particles dispersed in a suspension. DLS was used to determine the diffusion coefficient related to particle size according to the Stokes–Einstein equation (Equation (1)) that calculates the average *D_h_* of particles [64]. The *D_h_* is defined as the diameter of a hypothetical sphere that diffuses at the same rate as the particle, therefore including the hydration/solvation sphere surrounding the particle [65,66].
(1)$$D_{h}=\frac{kT}{3\pi ɳD_{T}}$$
where *k*: Boltzmann’s constant, *T*: Temperature, and ɳ: medium’s viscosity.

Before the heteroaggregation experiments, MPs and NPs were analyzed individually to assess their behavior in the different solvents. To this end, in a plastic (PMMA, Poly(methyl methacrylate)) disposable cuvette, 100 µL of MP or 200 µL of NP stock solutions was placed jointly with 900 µL or 800 µL in each solvent, respectively, to achieve a final concentration of 0.02 mg/mL of particles. DLS and ζ-potential measurements were conducted after 3 min sonication (at a frequency of 35 kHz) of each solution at 35 °C in an ultrasonic bath (J-P Selecta Ultrasounds HD, Barcelona, Spain). Subsequently, considering that MP and NP encapsulation by FIL is the first step towards the heteroaggregation process, two concentrations above the third CAC of FIL (69.22 and 132.15 mg/mL) [63] were selected by adding 50 µL and 100 µL of FIL stock to each particle solution (0.02 mg/mL), and then analyzed after overnight incubation at room temperature (r.t.). Afterward, through summative additions, different concentrations of chitosan (0.005, 0.019, and 0.025 µg/mL) were tested by adding 20, 75, and 100 µL of chitosan stock solution in each MP and FIL or NP and FIL dispersions. After each addition, a 5 min incubation at r.t. was carried out before DLS and ζ-potential measurements. Moreover, to further understand the interaction between chitosan and MPs or NPs, additional experiments with equivalent concentrations of chitosan (0.006, 0.021, and 0.027 µg/mL) were performed. It is worth mentioning that these procedures were initially performed in Milli-Q water, and then the synthetic seawater tests were carried out. For the reliability and validity of the results, measurements on each sample were performed at least in triplicate. The *D_h_* and ζ-potential results are expressed in nm (Z-average) and mV (ζ-potential), respectively.

#### 2.3.3. Scanning Electron Microscopy (SEM)

SEM provides morphological images in the microscopic range through a beam of electrons that react with atoms on the surface of a sample, leading to the emission of electrons from the surface. While scanning the surface, variations in the signals of the backscattered electrons occur according to the surface topology, thus providing information on the sample surface morphology and composition [67]. This technique was used for MP shape characterization in Milli-Q water and synthetic seawater. For SEM analysis, MP and NP solutions with concentrations of 0.02 mg/mL (dispersed in Milli-Q water and synthetic seawater) were applied on a carbon-coated adhesive, dried at room temperature, and examined using a scanning electron microscope (Auriga microscope, Carl Zeiss, Jena, Germany) with a 5 kV field and aperture size of 30 microns.

The experimental design and statistical analysis are shown in Appendix B.

## 3. Results 

### 3.1. Characterization of MP and NP Aggregation Method in Aquatic Systems

#### Microplastic and Nanoplastic Characterization

Before conducting the aggregation method experiments, the average size and surface morphology of polystyrene MPs (Figure 1(1)) and the amine-modified polystyrene NPs (PS-NH_2_ NPs) (Figure 1(2)) were determined in filtered Milli-Q water (FMQ) and filtered synthetic seawater (FSSW) via DLS measurements and SEM imaging analysis.

SEM analysis confirmed the spherical shape of MPs, and DLS measurements showed an MP diameter of 1121.90 ± 62.11 nm (PDI = 0.20 ± 0.17) and 1153.78 ± 83.30 nm (PDI = 0.28 ± 0.18) in FMQ and FSSW, respectively. The variability showed by DLS size distribution plots and the SEM image of MPs in FSSW indicates that the diameter of the MPs in the suspension is not uniform, presenting particles ranging from 600 nm to 1900 nm. In respect to sample stability, MPs in FMQ revealed a ζ-potential of −3.90 ± 1.44 mV (µ = −0.000030 ± 0.000011 cm^2^/Vs), which was statistically similar to −1.88 ± 3.84 mV (µ = −0.000014 ± 0.000030 cm^2^/Vs) found in FSSW. Therefore, regardless of the tested medium, MPs remained in a stable dispersion. SEM analysis confirmed the spherical shape of NPs and the mean diameter size provided by the manufacturer. Regarding PS-NH_2_ NPs in FMQ, DLS measurements showed a Z-average of 63.66 ± 4.01 nm (PDI = 0.18 ± 0.11) and a ζ-potential of 27.42 ± 10.42 mV (µ = 0.000213 ± 0.000081 cm^2^/Vs), revealing that NPs were well dispersed and presented a cationic charge on their surface. On the other hand, a remarkable aggregation was found when PS-NH_2_ NPs were suspended in FSSW, presenting a Z-average of 276.44 ± 36.78 nm (PDI = 0.51 ± 0.17). Figure 1(2) D reveals two peaks; the first (from 50.53 nm to 193.48 nm) shows a *D_h_* of 93.02 nm with a frequency of 8%, and the second peak (from 218.60 nm to 2837.04 nm) shows a *D_h_* of 513.71 nm with a frequency of 7%. Regarding colloidal stability, the PS-NH_2_ NPs in FSSW had a mean ζ-potential of 0.10 ± 4.26 mV (µ = 0.000001 ± 0.000033 cm^2^/Vs), which indicates a change in the NP surface charge. Considering the size (Z-average) and the aggregation/dispersion state (ζ-potential) obtained, synthetic seawater, as the dispersion medium, promotes the aggregation of the NP particles. These samples were used as controls for the heteroaggregation experiments. Results from the heteroaggregation experiments in filtered Milli-Q water (FMQ) with PS-NH_2_ NPs and Chit, PS-NP, FIL, and PS-NP with FIL and Chit, are summarized in Table 1.

Chitosan, when added at 0.019 µg/mL, led to the formation of larger aggregates of 8322.68 ± 2123.15 nm (*p* = 0.012) in comparison with the heteroaggregates formed in a sample of NPs (0.0182 mg/mL) and FIL (132.15 mg/mL). Regarding colloidal stability, there was a significant increase (*p* = 0.001) in the ζ-potential of the sample with 0.019 µg/mL of Chit. However, the values remained similar (*p* > 0.05) to the sample, with 0.0182 mg/mL NPs and 132.15 mg/mL FIL.

The results show that the samples with 0.0170 mg/mL NPs, 123.71 mg/mL FIL, and 0.019 µg/mL Chit presented the highest Z-average, indicating the largest NP heteroaggregates. The ζ-potential showed the closest value to the isoelectric point, thus revealing a tendency for charge neutralization through coagulation of the different colloidal components dispersed in these concentrations. The sample with the highest concentration of FIL, which presented the largest NP heteroaggregates, was then used for the summative additions of chitosan.

Considering the results observed in ultrapure water (Milli-Q), the same proportions of reagents (NPs, FIL, and chitosan) were applied to filtered synthetic seawater, and the Z-average and ζ-potential were measured. All the results are summarized in Table 2.

The trend of PS-NH_2_ NPs to aggregate in FSSW was enhanced in the presence of FIL, which fostered a significant increase (*p* = 0.002) in the size of the aggregates in the suspension (1165.48 ± 85.76 nm) compared to the respective control (276.44 ± 36.78 nm). Although the anionic character of the FIL led to a decrease in the ζ-potential of the colloidal solution to a value of −1.56 ± 2.70 mV, no significant difference was found compared to the control (*p* = 0.778). 

Concerning the NP heteroaggregation trial in filtered synthetic seawater, the largest size of particle aggregates (30 times that of 50 nm PS-NH_2_ NPs) was found in the sample with 0.0170 mg/mL NPs, 123.71 mg/mL FIL, and 0.019 µg/mL Chit, which accordingly presented the ζ-potential closest to the isoelectric point (point zero charge). Therefore, the NP aggregation method was applied using these optimized conditions to promote aggregation (NPs, FIL, and chitosan) in a bioassay using mussels as a proof of concept.

### 3.2. Biochemical Analyzes

No mortality was registered throughout the MP bioassay. Regarding the NP bioassay, no mortality was observed in the ERC (0.01 µg/mL) assay, contrary to the highest concentration exposure (2.5 µg/mL), where after 21 days of 2.5 µg/mL exposure to NPs, only two mussels out of five (≤40%) survived this concentration. 

#### 3.2.1. Nanoplastics

##### Superoxide Dismutase (SOD)

The results of SOD (% inhibition) measured in the gills and digestive glands of *M. galloprovincialis* are presented in Figure 2.

Before the beginning of the bioassays, SOD (% inhibition) was measured (T0), presenting values of 75.79 ± 12.09% and 71.29 ± 16.76% for the gills and digestive glands of mussels exposed to NPs, respectively, and 67.98 ± 4.88% and 40.80 ± 27.28% for the gills and digestive glands of mussels exposed to NPs with the aggregation method, respectively. When these values were compared with those of the respective controls, no significant differences were found except for controls in the gills after 7 (*p* = 0.009) and 21 days (*p* = 0.047) of exposure to NPs.

Regarding the bioassays using 0.01 µg NPs/mL (NPs and NPs with the aggregation method), SOD (%inhibition) in both tissues showed a trend of decreasing after 14 days of exposure in comparison to controls. Yet, this decrease was only significant in mussels’ gills exposed to NPs (*p* = 0.015) and in the digestive glands of mussels from the aggregation method bioassay (*p* = 0.022).

In the 2.5 µg NPs/mL bioassays (NPs and NPs with the aggregation method) in both tissues, no significant differences (*p* > 0.05) in SOD activity were registered.

Regarding the FIL control bioassay (not shown in Figure 2), after 14 days of exposure, the average SOD (% inhibition) values obtained for the gills and digestive glands were 46.34 ± 16.56% and 46.53 ± 7.95%, respectively, and did not present any significant differences compared with untreated mussels.

##### Catalase (CAT)

The results of CAT activity measured in the gills and digestive glands of *M. galloprovincialis* are represented in Figure 3.

Before the beginning of the bioassays (T0), CAT activity was measured and presented values of 0.10 ± 0.04 and 0.21 ± 0.11 nmol/min/mg total protein for the gills and digestive glands of mussels exposed to NPs, respectively, and 0.06 ± 0.04 and 0.35 ± 0.30 nmol/min/mg total protein for the gills and digestive glands of mussels exposed to NPs with the aggregation method. When these values were compared with the respective controls, no significant differences were found except for controls in the gills after 14 (*p* = 0.009) and 21 days (*p* = 0.009) of exposure to NPs.

The highest CAT activity values were measured in mussels exposed to 0.01 µg NPs/mL after 14 days of exposure to NPs (for both tissues).

Regarding the gills of mussels exposed to NPs (Figure 3A), there was a significant increase in CAT activity in all exposure periods of 0.01 µg NPs/mL, reaching a maximum value of 1.43 ± 0.74 nmol/min/mg total protein. However, in the highest concentration of tested NPs (0.02 µg NPs/mL), despite the same increasing trend, no significant differences were found compared to controls.

Regarding the FIL control bioassay (not represented in Figure 3), after 14 days, the average CAT activity values obtained in the gills and digestive glands were 8.87 ± 7.95 nmol/min/mg total protein and 14.86 ± 5.51 nmol/min/mg total protein, respectively, showing no significant differences compared with the control mussels.

##### Glutathione-S-Transferase (GST) 

GST activity was measured in the gills and digestive glands of *M. galloprovincialis*, and the results are shown in Figure 4.

Before the beginning of the bioassays (T0), GST activity was measured, presenting values of 134.33 ± 30.31 and 144.62 ± 114.24 nmol/min/mg total protein in the gills and digestive glands, respectively, in mussels exposed to NPs, and 41.10 ± 18.81 and 40.99 ± 15.49 nmol/min/mg total protein in the gills and digestive glands of mussels exposed to NPs using the aggregation method. When these values were compared with those of the respective controls, significant differences were found (*p* < 0.05), except in the controls in the gills after 7 days of exposure and in the digestive glands after 7 and 14 days of exposure to NPs.

The highest GST activity (349.28 ± 137.86 nmol/min/mg total protein) was found in the gills of mussels exposed to 0.01 µg/mL of NPs after 7 days, and the lowest value (23.61 nmol/min/mg total protein) was detected in the gills of mussels exposed to 2.5 µg/mL of NPs with the aggregation method. Concerning mussels exposed to NPs, at 0.01 µg NPs/mL, a significant increase in GST activity was found after 7 and 14 days of exposure in the gills (*p* = 0.016; *p* = 0.014) and after 7 days of exposure in the digestive glands (*p* = 0.056). GST activity was lower in the gills of mussels exposed to 2.5 µg/mL of NPs compared to those of controls. On the other hand, in the digestive glands of mussels exposed to 2.5 µg/mL of NPs, the GST values were higher compared to those of the respective controls. However, for both tissues, no significant differences were found (*p* > 0.05). Moreover, compared to controls, no significant differences (*p* > 0.05) were found in both tissues and NP concentrations for mussels exposed to NPs with the aggregation method. When treated mussels in both bioassays were compared, the GST activity was lower when exposed to NPs with the aggregation method. Regarding the FIL control bioassay (not represented in Figure 4), after 14 days, the average GST activity values obtained in the gills and digestive glands were 39.63 ± 7.74 nmol/min/mg total protein and 28.06 ± 15 nmol/min/mg total protein, respectively. The digestive glands showed significant differences compared with untreated mussels (*p* = 0.034).

##### Total Antioxidant Capacity (TAC) 

TAC was measured in the gills and digestive glands of *M. galloprovincialis*, and the results are presented in Figure 5. 

Before the bioassays (T0) began, TAC was measured, showing values of 0.03 ± 0.01 and 0.08 ± 0.05 nmol/min/mg total protein for the gills and digestive glands of mussels exposed to NPs, respectively. TAC values were 0.03 ± 0.03 and 0.07 ± 0.05 nmol/min/mg total protein in mussels’ gills and digestive glands exposed to NPs with the aggregation method, respectively. When T0 values were compared to those of controls, no significant differences were found in the NP test except controls in the gills after 21 days of exposure (*p* = 0.009), whereas, in the NP bioassay with the aggregation method, there were significant differences (*p* < 0.05) among all the different tissues and exposure periods.

Regarding the NP bioassay, TAC values in the gills ranged from 0.03 ± 0.02 nmol/min/mg total protein (control after 7 days of exposure) to 0.54 ± 0.33 nmol/min/mg total protein (0.01 µg NPs/mL after 14 days of exposure). In the digestive glands, the lowest TAC value was 0.05 ± 0.04 nmol/min/mg total protein (control after 21 days of exposure), and the highest was 0.59 ± 0.19 nmol/min/mg total protein (0.01 µg NPs/mL after 14 days of exposure).

After 7 and 14 days of exposure to NPs, both tissues exposed to 0.01 µg NPs/mL showed a significant increase in TAC (*p* = 0.01 and *p* = 0.03 for the gills; *p* = 0.05 and *p* = 0.01 for the digestive glands, respectively; Figure 5A,C) compared to controls.

No significant differences were found in 2.5 µg/mL of NP exposure; however, regardless of the tissue and exposure period, there was a trend for TAC to increase compared to controls.

Concerning the NP bioassay with the aggregation method, the lowest TAC was found in 2.5 µg NPs/mL exposure trials, showing values of 0.05 nmol/min/mg total protein and 0.07 nmol/min/mg total protein for the gills and digestive glands, respectively. In both tissues of mussels exposed to 0.01 µg NPs/mL, no significant differences were found compared to controls.

Moreover, regarding the FIL control bioassay (not represented in Figure 5), after 14 days, the average TAC values obtained in the gills and digestive glands were 0.03 ± 0.02 nmol/min/mg total protein and 0.11 ± 0.13 nmol/min/mg total protein, respectively, and did not show significant differences compared to control mussels.

##### Glutathione Peroxidase Activity (GPx) 

GPx activity was measured in the gills and digestive glands of *M. galloprovincialis*, and the results are represented in Figure 6.

Before the beginning of the bioassays (T0), GPx activity was measured and presented values of 10.67 ± 5.84 and 19.74 ± 16.41 nmol/min/mg total protein for the gills and digestive glands of mussels exposed to NPs, respectively, and 12.92 ± 7.07 and 14.62 ± 5.72 nmol/min/mg total protein for the gills and digestive glands of mussels exposed to NPs with the aggregation method, respectively. No significant differences were found when compared to the respective controls.

The highest GPx activity (34.34 nmol/min/mg total protein) was found in the gills of mussels exposed to 0.01 µg/mL of NPs after 14 days of exposure, and the lowest (3.6 nmol/min/mg total protein) in the mussels exposed to 2.5 µg/mL of NPs after 7 days of exposure.

A significant increase was registered in the gills of mussels exposed to 0.01 µg/mL of NPs after 14 and 21 days of exposure (*p* = 0.022 and *p* = 0.030). In contrast, no significant differences were found in the highest exposure concentration compared to the controls.

Regarding the digestive glands of mussels treated with NPs, when compared to the controls, there were significant differences, except for 0.01 µg/mL after 21 days of exposure (*p* = 0.042). Moreover, no significant differences (*p* > 0.05) were found in both tissues and NP concentrations in mussels exposed to NPs with the aggregation method compared to controls. Regarding the FIL control bioassay (not shown in Figure 6), after 14 days of exposure, the average GPx activity values obtained in the gills and digestive glands were 8.87 ± 1.02 nmol/min/mg total protein and 14.86 ± 5.51 nmol/min/mg total protein, respectively, and did not show significant differences compared to control mussels.

##### Lipoperoxidation (MDA Content)

Lipoperoxidation was measured in the gills and digestive glands of *M. galloprovincialis*; the results are expressed in terms of MDA concentration and presented in Figure 7.

Before the beginning of the bioassays (T0), LPO presented MDA concentration values of 1.97 ± 1.47 nmol/min/mg total protein in the gills and 3.61 ± 2.47 nmol/min/mg total protein in the digestive glands exposed to NPs, respectively, and 5.65 ± 2.19 and 4.39 ± 2.1 nmol/min/mg total protein in the gills and digestive glands of mussels exposed to NPs using the aggregation method. The statistical analysis showed no significant differences in the gills after 14 days of exposure (*p* = 0.028) and in the digestive glands after 7 days (*p* = 0.009) and 14 days of exposure to NPs with the aggregation method compared to the respective controls.

Regarding the exposure to NPs, a significant increase in LPO was registered in the gills of mussels exposed to 0.01 µg NPs/mL for all exposure periods, while in the digestive glands, this increase was only significant after 21 days of exposure compared to controls.

Although no significant differences (*p* > 0.05) were found compared to controls, generally, after 7 days of exposure to 0.01 µg NPs/mL using the aggregation method, the MDA values increased in the gills and digestive glands.

In both NP bioassays (with and without the aggregation method), the highest MDA concentrations were found in the gills at 0.01 µg NPs/mL.

Regarding the FIL control bioassay (not shown in Figure 7), after 14 days of exposure, the average MDA concentration values determined in the gills and digestive glands were 3.58 ± 1.29 nmol/min/mg total protein and 3.63 ± 2.49 nmol/min/mg total protein, respectively. Still, no significant differences were found compared to control mussels.

##### Total Ubiquitin (Ub)

Total ubiquitin concentration was measured in the gills and digestive glands of *M. galloprovincialis*, the results represented in Figure 8.

Before the beginning of the bioassays (T0), total ubiquitin was measured and presented average concentrations of 0.014 ± 0.014 nmol/min/mg total protein in the gills and 0.0031 ± 0.0024 nmol/min/mg total protein in the digestive glands of mussels exposed to NPs, and 0.0015 ± 0.002 and 0.0072 ± 0.0064 nmol/min/mg total protein in the gills and digestive glands of mussels exposed to NPs with aggregation method, respectively. These values showed significant differences in mussel gills and digestive glands after 14 days of exposure (*p* < 0.05) to NPs and in all tissues and exposure times in the bioassay of NPs with the aggregation method, compared to the respective controls.

The highest ubiquitin concentration (0.13 ± 0.05 nmol/min/mg total protein) was found in the gills of mussels exposed to 0.01 µg NPs/mL of NPs after 14 days of exposure.

In both the gills and digestive glands of mussels exposed to 0.01 µg NPs/mL after 7 days of exposure, there was a significant increase in total ubiquitin compared to controls.

Regarding the bioassay with the aggregation method, no significant differences were found. Yet a noteworthy decrease in the total ubiquitin concentration was registered in both tissues of mussels exposed to 2.5 µg NPs/mL. Moreover, in general, mussels exposed to NPs using the aggregation method showed lower ubiquitin concentration values than mussels exposed to NPs.

Concerning the FIL control bioassay (not represented in Figure 8), after 14 days of exposure, the average ubiquitin concentration in the gills and digestive glands was 8.26 ± 2.85 nmol/min/mg total protein and 0.96 ± 0.78 nmol/min/mg total protein, respectively, showing significant differences compared to the gills of control mussels (*p* = 0.02).

To assess the presence of MPs and NPs in mussel’s tissues, a histological analysis was carried out. Figure 9 and Figure 10 present the results from the histological observations of the gills of mussels exposed to the highest concentrations of plain PS MPs (0.02 µg/mL) and PS-NH_2_ NPs using the aggregation method (2.5 µg/mL), respectively.

Results from the microscope observations of MPs confirm the spherical form of MPs, which presents a light blue color (Figure 9A). These 1 µm particles were not found in the gills of mussels after 21 days of exposure to 0.02 µg MPs/mL (Figure 9B). On the other hand, NP aggregates were observed at the surface of mussels’ gills (near cilia) after 7 days of exposure to NPs using the aggregation method (Figure 10B). Generally, gills present a normal appearance with no evidence of MP (Figure 9B) or NP aggregates (Figure 10B) inside tissues.

## 4. Discussion

Plastics’ remarkable properties and economic value make them ubiquitous in today’s society. Many of these plastics are not recycled, and due to their mismanagement, tonnes of these materials are discarded in the oceans yearly. Constant weathering of plastic in the ocean leads to its fragmentation into micro- or even nanoplastics (MNPs). These materials’ microscopic and nanoscopic sizes pose a great challenge to their detection and capture from the environment [68]. Thus, in the present work, a method is proposed to promote the aggregation of MNPs to facilitate, by increasing the litter’s size, their detection and extraction from the marine ecosystem. Polystyrene (PS) was selected as a model since its products are commonly used daily, mainly for one single-use purpose, and, as reported, are frequently detected in the environment and can acquire micro- and nanoscopic sizes [3,6,69]. Moreover, plain PS MPs were used for the microplastics bioassay, whereas amino-modified PS NPs (PS-NH_2_) were used for the nanoplastics bioassay since hindered amine light stabilizers (HALS) are one of the most used UV stabilizers in plastics [2,4].

This aggregation method is based on ionic-liquid systems that self-assemble into micelle-shaped structures and chitosan (Chit). Chitosan is a polycationic polymer derived from the deamination of chitin, which is a structural element of the crustaceous exoskeleton [70,71], and was chosen to act as a natural polyelectrolyte that is expected to promote the further increase in the MNP heteroaggregates. In that regard, a fluorinated ionic liquid (FIL) was chosen, presenting critical aggregation concentrations (CACs) similar to those of perfluorinated anionic surfactants. In this study, two concentrations above the third CAC of FIL were selected. Pereiro et al. [63] reported that FIL self-assembling structures present a cylindrical or lamellar micelle shape in these concentrations. Therefore, according to our hypothesis, due to the intrinsic hydrophobicity of PS, it is predicted that PS MNPs will be encapsulated by the FIL self-assembling micelle structures, leading to the formation of MNP heteroaggregates [63,72,73]. To assess particle size and stability, a characterization process of the aggregation method was carried out by DLS and ζ-potential measurements, which are frequently used in MNP studies [27,62,74]. Firstly, before performing the characterization of the aggregation method, PS MP and PS-NH_2_ NP behaviors in aqueous solutions were evaluated. The results showed that particle surface charge determines MNP interaction with the surrounding medium. When plain PS MPs and PS-NH_2_ NPs were suspended in synthetic seawater, the MPs were revealed to be neutral and maintained their dispersed state by not displaying any surface charge. In contrast, the surface charge of the cationic NPs changed drastically and rapidly formed NP aggregates that were 5-fold their original size [75]. PS NP’s tendency to aggregate in seawater had been previously reported by Wegner et al. [76] and Brandts et al. [62], and explained by several authors [17,25,77]. According to DLVO theory, the solvent’s ionic strength and electrolyte content significantly influence colloidal stability and the particle aggregation state. Hence, first, electrolytes like Cl^−^, Br^−^ and SO_4_^2−^ dispersed in the synthetic seawater led to particle charge reversal through adsorption onto the positive charge (NH_2_) surface of PS NPs, resulting in the decrease in interparticle electrostatic repulsion. Subsequently, additional attractive forces, triggered by divalent cations such as Mg^2+^, Sr^2+^, and Ca^2+^, enabled NP heteroaggregation via the bridging effect [23,26]. Furthermore, the ζ-potential of NPs significantly changed in seawater (0.10 ± 4.26 mV) compared with ultrapure water (27.42 ± 10.42 mV), which corroborates the ions’ adsorption onto the NP surface. These findings are in agreement with the results of Brandts et al. [62] and Canesi et al. [74].

Regarding the application of the aggregation method, concerning MPs, none of the elements (Chit and FIL) led to changes in particle size; however, upon the addition of FIL, the ζ-potential and electrophoretic mobility decreased. The results suggest that FIL adsorption onto the PS MP surface occurred due to hydrophobic attraction [73]. However, with the increase in FIL concentration, it was observed that the ζ-potential became less negative, suggesting that fewer FIL ions were adsorbed on the PS MP surface. In general, no trends were found regarding the MP behavior with the technology components, either alone or combined; therefore, no conclusive characterization can be inferred.

On the other hand, in the NP heteroaggregation experiments in FMQ, FIL and Chit. were shown to be valuable tools to promote the heteroaggregation of PS-NH_2_ NPs, demonstrating remarkable increases in the *D_h_* of the particles. The ζ-potential of NPs drastically shifted from positive to negative, indicating that the FIL ions rapidly coated the particles. As described by Jódar-Reyes et al. [78] and González et al. [79], the hydrophobic attraction associated with the electrostatic bond between the anionic tail of the surfactant and the cationic surface of particles enabled the formation of a surfactant–polymer complex. Afterwards, chitosan (0.019 µg/mL) bridges between FIL and NP complexes allowed a further increase in the heteroaggregates in the suspension [27]. This process resulted in a significant increase in the ζ-potential, indicating that charge neutralization of the NP heteroaggregates occurred. It was found that the maximum heteroaggregation was achieved when the ζ-potential values were closer to zero. Accordingly, previous studies have reported that charge neutralization at the isoelectric point determines the formation of larger NP heteroaggregates [26]. Furthermore, as expected, no aggregates were formed when chitosan was added alone, since both chitosan and NPs exhibited the same charge, further increasing interparticle repulsion. However, no conclusive results were obtained regarding ζ-potential. In general, these results suggest that above the third CAC, FIL self-assembling micelle-shaped structures efficiently encapsulated PS-NH_2_ NPs, leading to the formation of NP heteroaggregates with micrometric sizes (4552.53 ± 613.68 nm). Moreover, the largest NP heteroaggregates (8322.68 ± 2123.15 nm) were obtained in the highest concentration of FIL, with 0.019 µg/mL of chitosan. Therefore, the same proportions of the reagents (NPs, FIL, and chitosan) were applied in filtered synthetic seawater. The results showed that when FIL was applied to NPs in synthetic seawater, despite having contributed to the formation of NP heteroaggregates, the aggregates obtained were considerably smaller (1595.27 ± 672.54 nm) compared to those found in the FMQ experiment. The adsorption of synthetic seawater (SSW) ions onto the PS NP surface, formerly described, may have decreased the reactivity of PS-NH_2_ NPs towards the FIL since there was a decrease in the surface area and charge neutralization of particles [80,81]. Therefore, the surface charge of the particles became less positive, decreasing the affinity and, consequently, the adsorption of the FIL, which ultimately led to the formation of smaller heteroaggregates compared to the FMQ experiment [82]. Nevertheless, overall, the results demonstrated that this aggregation method successfully enabled the transition from the nanometric to the micrometric scale of the PS-NH_2_ NPs.

MPs are a new niche in the marine environment, and several species of aquatic biota are vulnerable to this type of contaminant, thus threatening marine biodiversity [68]. For this reason, to evaluate the toxic potential of MNPs, *M. galloprovincialis* was exposed to polystyrene MPs and NPs separately. Moreover, as a proof of concept for the aggregation method developed, the mussels were exposed to PS-NH_2_ NPs using the aggregation method, and the toxicity of these contaminants was determined.

The uptake of MNPs from water by mussels has been demonstrated in various studies [83,84,85]. Despite plastics having been reported as inert materials [43], MNP internalization has been shown to induce changes in the feeding, filtration behavior, and energy consumption of these organisms [76,86]. Moreover, it has been reported that MPs and NPs trigger the production of ROS in mussels [74,87]. ROS production can induce oxidative stress. Oxidative stress can be measured by quantifying ROS, the activity of antioxidant defenses, or oxidative damage [88,89]. The activity of antioxidant enzymes can be enhanced upon induced stress by xenobiotics. Yet if stress persists, the defense mechanisms can be compromised, so the inhibition of the enzymatic activities can occur [55]. Ultimately, this can lead to the organism’s exhaustion, causing it to become more susceptible to environmental stressors [90].

Therefore, in this study, a multi-biomarker approach to assess the redox homeostasis of *M. galloprovincialis* exposed to MNPs was carried out. Antioxidant enzyme activities (SOD, CAT, GST, and GPx), radical scavenging capacity (TAC), and cellular (LPO) and protein damage (Ub) were analyzed. In general, results suggest that long-term exposure (21 days) to PS MPs causes oxidative stress in *M. galloprovincialis.* Concerning NP exposure tests, mussels exposed to the ERC showed oxidative stress, but at this NP concentration, the organisms could develop efficient compensatory mechanisms. Furthermore, applying the aggregation method proved valuable for decreasing NPs’ bioavailability and toxicity at the ERC. Distinct responses between gills and digestive glands were observed. This can be attributed to both organs’ different physiological functions and cellular differences [91]. For instance, in the NP trials, gills were the tissues most affected, showing the highest activity values for all biomarkers except TAC.

SOD is an enzyme that regulates the levels of superoxide anions (O_2_^•−^) in the organisms [88]. This important enzyme acts in the first line of antioxidant defense systems of bivalves [92]. The results reveal that, overall, regardless of the tissue, SOD activity remained similar to controls for all NP concentrations (whether with or without the aggregation method) and exposure periods. Therefore, we postulate that SOD is not the antioxidant defense mechanism used by *M. galloprovincialis* to respond to PS NP toxicity. Similar results were found by Li et al. [92] in the gills of *C. fluminea* after 4 days of exposure to 0.1 and 1 mg/L of PS NPs.

Catalase is an enzyme that neutralizes hydrogen peroxide, converting this radical into water [93]. Regarding CAT activity in the NP bioassays, a significant increase in CAT activity was found in the gills and digestive glands of mussels exposed to the ERC of NPs after 14 and 21 days of exposure compared to controls. Despite not being significant, the same increasing trend was registered in the highest concentration of NPs. When mussels were exposed to the ERC of NPs using the aggregation method in both tissues, there was a significant increase in CAT activity after 7 days of exposure compared to controls. These results reveal that CAT activity was stimulated to cope with the ROS production triggered by PS NPs [90]. Yet, it is worth mentioning that when NPs were applied with the aggregation method of the ERC, after 14 days of exposure, CAT values returned to normal, compared to controls, indicating that the oxidative challenge was short-lived [94].

GST is an enzyme responsible for the detoxification of toxic xenobiotics through the catalysis of conjugated compounds with higher solubility, less toxicity, and ultimately facilitating their excretion [95,96,97]. Moreover, GST enzymes contribute to the breakdown of lipid peroxides, preventing membrane damage [95]. Relative to the NP bioassays, GST activity significantly increased in the gills after 7 and 14 days of exposure to the ERC of NPs, compared to controls. In the digestive glands, in both NP concentrations, GST activity was higher after 7 days of exposure compared to controls, which over time tended to decrease. The increase in GST activity is supported by Li et al. [92], who found higher GST values in the gills of *C. fluminea* after exposure to 80 nm PS NPs. The following depletion of GST activity can be associated with the gradual exhaustion of the organisms throughout the ROS overproduction induced by PS NP exposure [55]. In the NPs with the exposure of the aggregation method, regardless of the tissue, GST activity remained constant compared to controls, which suggests that the method positively contributed to hindering the adverse effects of oxidative stress induced by NPs.

Total antioxidant capacity (TAC) is a biomarker commonly used to measure the overall response capacity against oxidative stress [98]. Concerning NP exposure trials, regardless of the mussel’s tissue, TAC values significantly increased after 7 and 14 days of exposure to an ERC of NPs compared to controls. Despite not being significant, the same trend was found in both tissues at the highest NP concentration. Brandts et al. [62] also showed increased TAC when *M. galloprovincialis* was exposed to high concentrations (50 mg/L) of PS NPs. On the other hand, after 21 days of exposure, TAC values decreased in both tissues when mussels were exposed to NPs at an ERC. Therefore, our results suggest that the organism’s capacity to scavenge free radicals was compromised over time due to an overwhelming accumulation of oxyradicals [55]. Regarding the bioassay of NPs using the aggregation method, in the ERC, no significant differences in TAC values were registered, whereas in the TRC, a noteworthy decrease was observed after 7 days of exposure in both studied tissues. These results suggest that when the aggregation method is applied at low concentrations, NPs are prevented from inducing oxidative stress in the organism. However, in addition to NPs, high concentrations of FIL probably enhance the production of ROS, and, therefore, mussels could not positively develop compensatory antioxidant mechanisms of defense. Moreover, the deaths observed in the control exposure of the highest FIL concentration seem to confirm this chemical’s toxicity potential.

GPx is an enzyme that breaks down hydrogen peroxide and inactivates metabolites from lipid peroxidation, such as MDA [99,100]. A significant increase in GPx activity was found in gills after 14 and 21 days of exposure to NPs at the ERC and in the digestive glands after 21 days of exposure, compared to controls. These results are similar to CAT, and considering that both enzymes participate in the decomposition of H_2_O_2_, we postulate that this ROS may have been extensively produced over time, requiring the coactivation of GPx and CAT enzymes. Moreover, bearing in mind that GSH is a common substrate for GPx and GST, this enhancement in GPx activity led to increased GSH consumption, contributing to GST depletion after 21 days of exposure. In contrast, at the TRC, NP exposure did not induce significant changes in GPx activity in both tissues compared to controls. Furthermore, in the NPs with the aggregation method bioassay, the GPx values did not show significant differences compared to controls.

Lipid peroxidation is a common consequence of ROS accumulation in living organisms [89]. MDA is frequently used as a lipid peroxidation biomarker, considering that it is one of the final products of the peroxidative reaction of lipids [55,101].

Oxidative damage was also observed in the NP trials. Both tissues revealed a significant increase in lipoperoxidation after 21 days of exposure to NPs at an ERC compared to controls. However, the gills were more sensitive to ROS since a significant increase in LPO was also found after 7 and 14 days of NP exposure at the ERC compared to controls. It can be explained considering that, besides the important role of gills in detoxification processes, this is one of the first organs in contact with contaminants, in this case, NPs, present in water, presenting early oxidative damages [62]. An increasing trend of LPO was also registered in the TRC in both gills and digestive glands, but the values were not significantly higher than those of the controls. Our results agree with the findings of Brandts et al. [62], evidencing the same increasing trend in LPO in the gills and digestive glands of *M. galloprovincialis* after exposure to PS NPs (0.05 µg/mL). Compared to NP trials, when the aggregation method was applied with the NPs, LPO showed no significant differences compared to controls, suggesting that the aggregation method limited the oxidative degradation of lipids caused by the NPs.

The total ubiquitin (Ub) assay was carried out to evaluate the protein quality and integrity since the increase in ROS can lead to the formation of malfunctioning oxidized proteins, which become more susceptible to removal and degradation [88,102]. Moreover, ubiquitin-dependent proteasomal degradation machinery is sensitive to oxidative stress, and begins to reduce its activity in high H_2_O_2_ concentrations [103]. Therefore, under oxidative stress, an increase in the total ubiquitin protein is expected to signal damaged proteins for degradation [102,103]. The NP trial gills and digestive glands showed a significant increase in the total concentration of ubiquitin after 7 days of exposure to the ERC of NPs compared to controls. Moreover, when mussels were exposed to the TRC of NPs, both tissues showed an increasing trend in total ubiquitin at all exposure periods, compared to controls. It has been previously reported that nanoplastics can lead to protein misfolding [104], possibly activating the ubiquitin-mediated protein degradation pathway and increasing the total ubiquitin in the organisms. Notwithstanding this, when mussels were exposed to NPs with the aggregation method at the TRC, a decrease in total ubiquitin was found after 7 days of exposure, indicating that the aggregation method components, particularly the ionic liquid, are probably toxic and may have created additional stress in the organisms. Under persistent oxidative stress, the ubiquitination system is inactivated, and the ubiquitin levels tend to decrease [88]. Instead, in these cases, proteasome machinery, independent of ubiquitin, that requires less energy and is less sensitive to oxidative stress, is more active and ensures the proper degradation of damaged proteins [103]. According to Wegner et al. [76], mussels tend to decrease their filtering activity when exposed to PS NPs; therefore, by filtering lower volumes of water, the mussel’s feeding capability may be compromised, eventually resulting in an energetic deficiency [1]. Hence, it is hypothesized that malnutrition may have contributed to this bioassay’s long-term (21 days) deaths. However, when mussels were exposed to NPs at the TRC with the aggregation method, a significant mortality rate was recorded (approximately 90% out of the total), and the only mussel that remained alive showed depletion in TAC and ubiquitin content after 7 days of exposure. Moreover, concerning FIL control trials, high concentrations (18.19 mg/mL) of FIL were demonstrated to be highly toxic to mussels since no organism survived after 7 days of exposure. Nevertheless, the histological observation did not reveal tissue damage or the presence of NPs within the mussel gills exposed to NPs with the aggregation method at the TRC. Therefore, despite the results suggesting that FIL decreased NP bioavailability by promoting their aggregation, this chemical can be considered harmful to *M. galloprovincialis*. More toxicological tests need to be conducted in seawater for a more accurate hazard assessment of FIL in marine organisms.

Overall, these results suggest that mussels are more sensitive to NPs than MPs (please see Appendix A) since significant variations in the antioxidant enzymatic activities, radical scavenging capacity (TAC), and cellular (LPO) and protein damage were registered earlier in the NP trials compared to the MP trials for the same ERC.

## 5. Conclusions

One of the most challenging aspects of detecting and capturing MNPs from the environment is related to their scale. The present study demonstrated that FIL is a promising tool to promote the aggregation of NPs. However, the surface charge of these colloidal particles played a pivotal role in the aggregation process, considering that an efficient aggregation of cationic polystyrene nanoplastics (Ps NP-NH_2_) was found. In contrast, plain PS MPs maintained their dispersive state. Toxicological studies of long-term exposure to MNPs are still scarce. However, they are of great importance since these types of pollutants are continually accumulating in the environment, and therefore, it may be more relevant to study chronic effects than acute effects. In this work, *M. galloprovincialis* were exposed for 21 days to MNPs. Data showed some variability in the activity of antioxidant enzymes—radical scavenging capacity, and cellular (LPO) and protein damages. The results suggest that impairment in the ROS homeostasis occurred in both MP and NP bioassays, although for MPs, this was only notable after 21 days of exposure. Moreover, results from the NP exposure trials suggest that the aggregation of NPs using this aggregation method effectively decreases the bioavailability of NPs at an ERC since, in these conditions, fewer adverse effects regarding oxidative stress were found in the exposed mussels. However, FIL showed toxicity when applied at high concentrations; therefore, an optimization process is being carried out to ensure that this aggregation method has a high yield coupled with low toxicity. This study allowed the development of more integrated knowledge of MNPs in the marine environment by linking physical (particle size and concentration), chemical (particle surface charge), and toxicological (particle exposure effects on oxidative responses) fields. Further work intends to create a more realistic model by exposing MNPs to actual environmental weathering conditions and thus performing more accurate exposure trials.

## Figures and Tables

**Figure 1 biomimetics-09-00303-f001:**
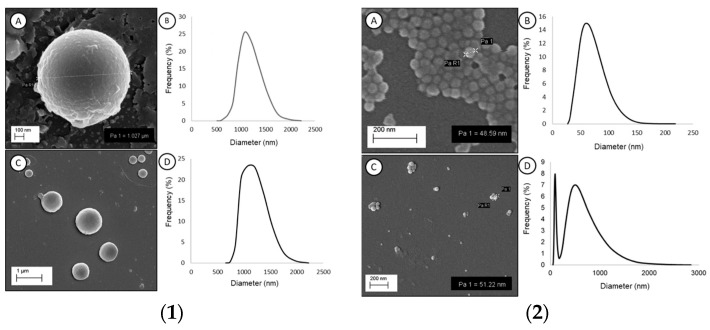
(**1**) Representative SEM images (A and C) and size distribution of polystyrene microplastics (B and D) in FMQ (A and B) and FSSW (C and D). Data from DLS are presented as total mean diameter. (**2**) Representative SEM images (A and C), size distribution (B and D), and size distribution of PS-NH_2_ NPs in FMQ (A and B) and FSSW (C and D). Data from DLS are expressed as total mean diameter.

**Figure 2 biomimetics-09-00303-f002:**
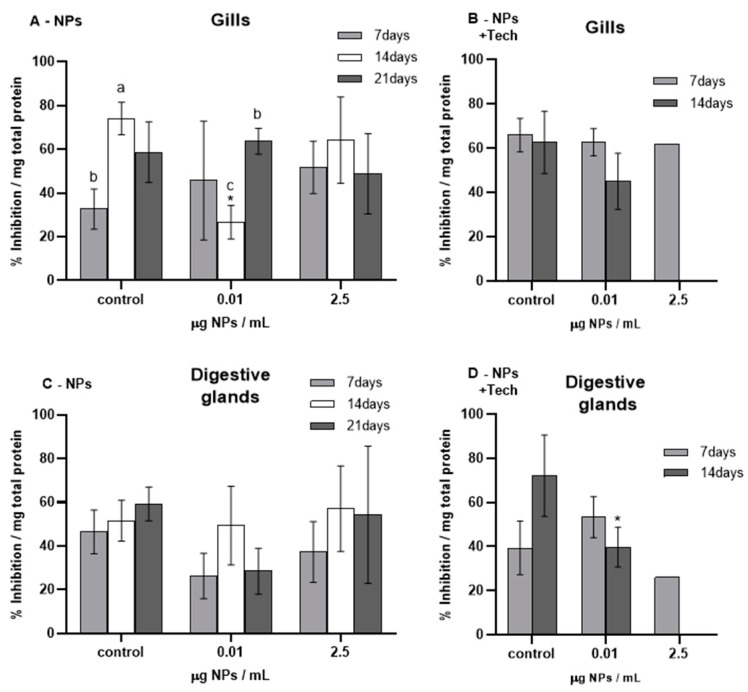
Percentage inhibition of SOD (mean ± sd) in *M. galloprovincialis* exposed to NPs and NPs using the aggregation method at different concentrations measured in gills (**A**,**B**) and digestive glands (**C**,**D**) after 7, 14, and 21 days of exposure. Significant differences (*p* < 0.05) compared to controls are denoted by *. Legend: lowercase letters a, b, and c indicate significant differences (*p* < 0.05) between 7, 14, and 21 days of exposure, respectively.

**Figure 3 biomimetics-09-00303-f003:**
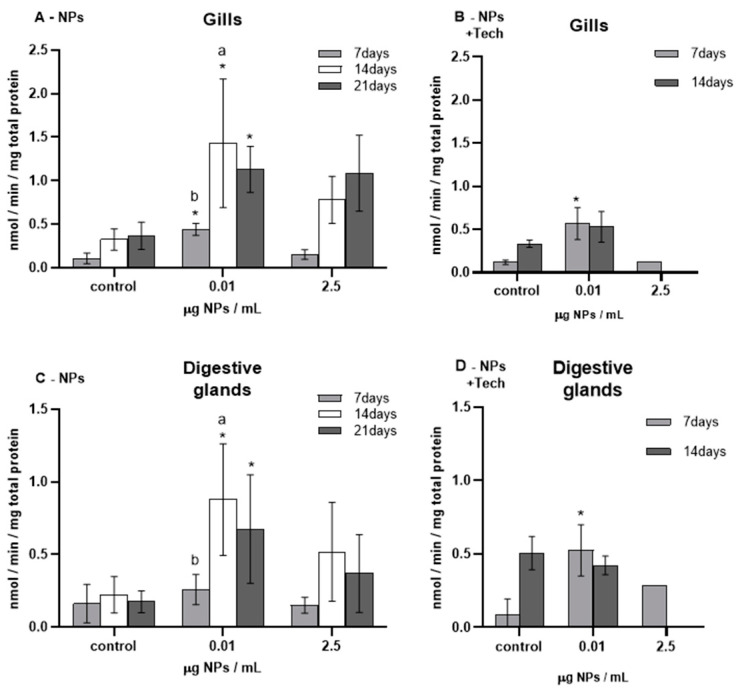
Catalase activity (mean ± sd) in *M. galloprovincialis* exposed to NPs and NPs using the aggregation method at different concentrations measured in gills (**A**,**B**) and digestive glands (**C**,**D**) after 7, 14, and 21 days of exposure. Significant differences (*p* < 0.05) compared to controls are denoted by *. Legend: lowercase letters a, and b indicate significant differences (*p* < 0.05) between 7, 14, and 21 days of exposure, respectively.

**Figure 4 biomimetics-09-00303-f004:**
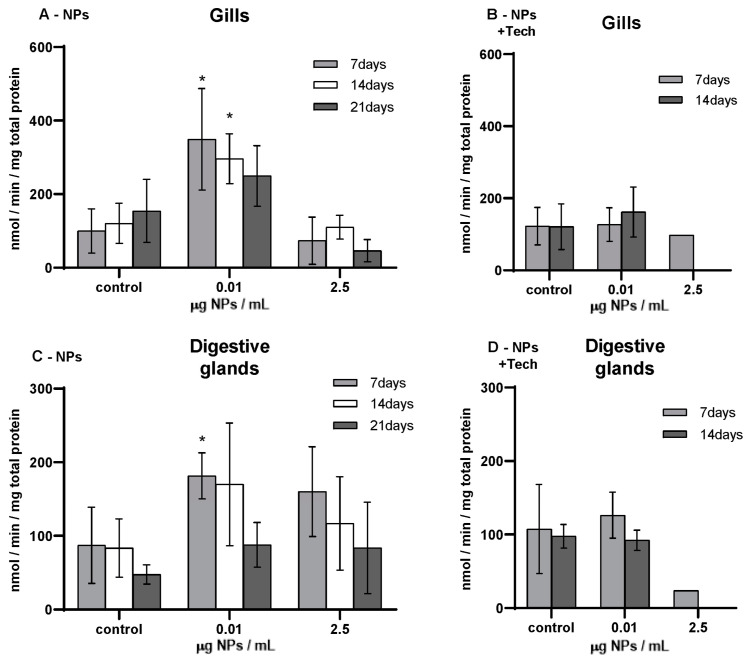
GST activity (mean ± sd) in *M. galloprovincialis* exposed to NPs and NPs using the aggregation method at different concentrations measured in gills (**A**,**B**) and digestive glands (**C**,**D**) after 7, 14, and 21 days of exposure. Significant differences (*p* < 0.05) compared to controls are denoted by *.

**Figure 5 biomimetics-09-00303-f005:**
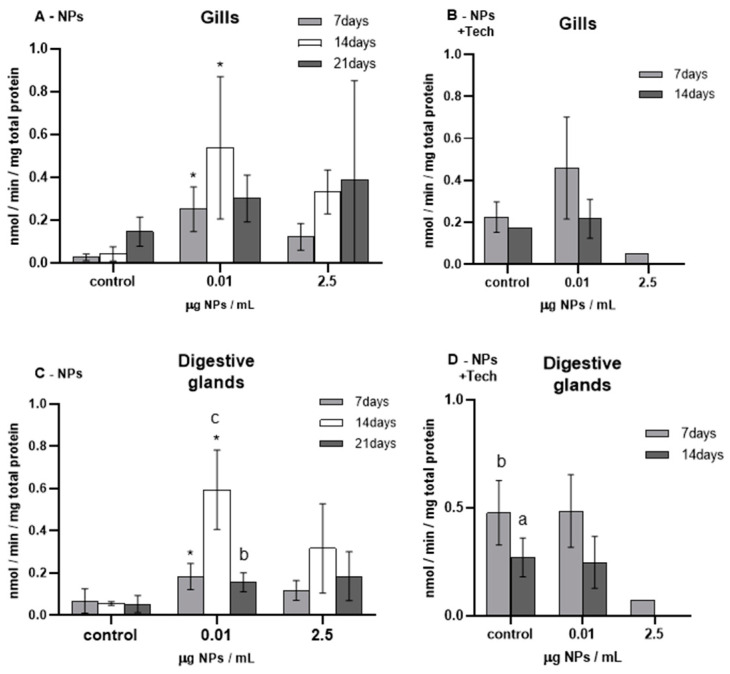
Total antioxidant capacity (mean ± sd) in *M. galloprovincialis* exposed to NPs and NPs using the aggregation method at different concentrations measured in gills (**A**,**B**) and digestive glands (**C**,**D**) after 7, 14, and 21 days of exposure. Significant differences (*p* < 0.05) compared to controls are denoted by *. Legend: lowercase letters a, b, and c indicate significant differences (*p* < 0.05) between 7, 14, and 21 days of exposure, respectively.

**Figure 6 biomimetics-09-00303-f006:**
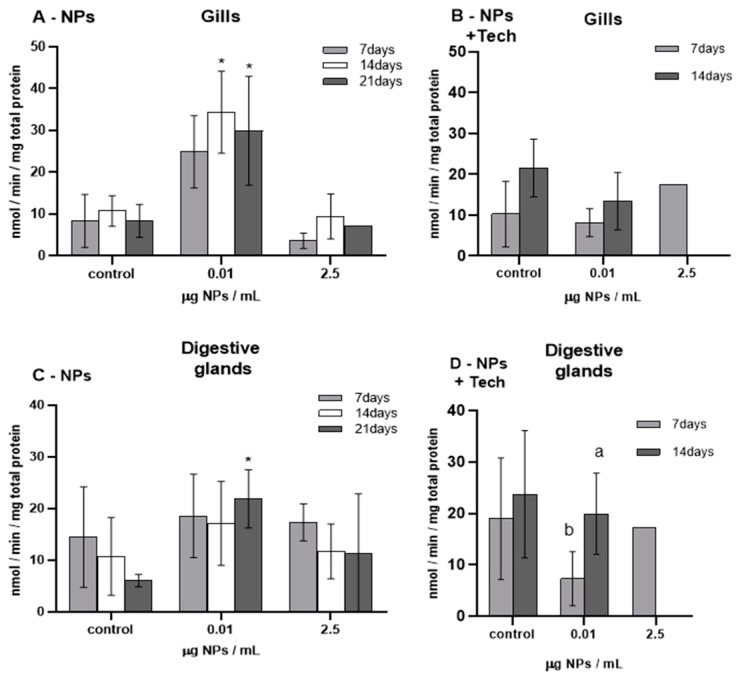
GPx activity (mean ± sd) in *M. galloprovincialis* exposed to NPs and NPs using the aggregation method at different concentrations measured in gills (**A**,**B**) and digestive glands (**C**,**D**) after 7, 14, and 21 days of exposure. Significant differences (*p* < 0.05) compared to controls are denoted by *. Legend: lowercase letters a, and b indicate significant differences (*p* < 0.05) between 7, 14, and 21 days of exposure, respectively.

**Figure 7 biomimetics-09-00303-f007:**
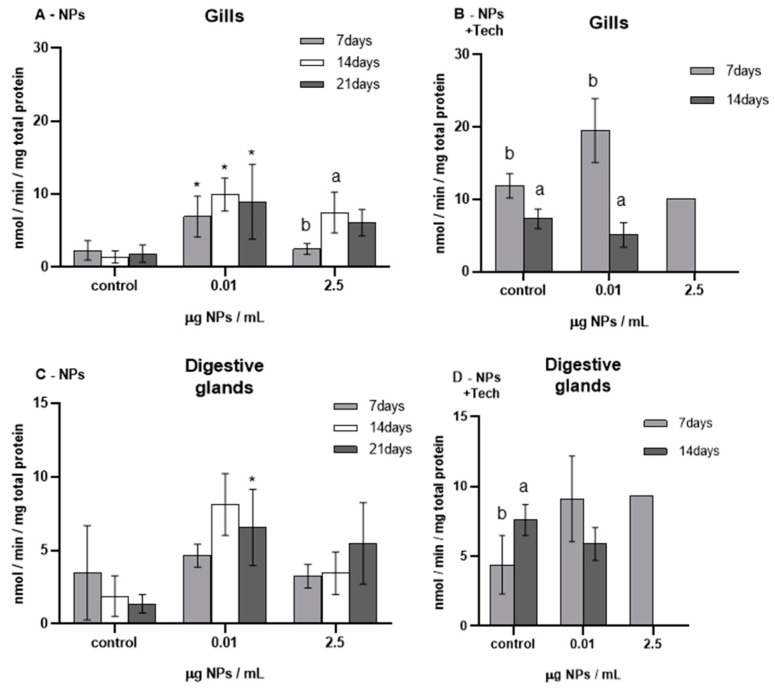
MDA concentration (mean ± sd) in *M. galloprovincialis* exposed to NPs and NPs using the aggregation method at different concentrations measured in gills (**A**,**B**) and digestive glands (**C**,**D**) after 7, 14, and 21 days of exposure. Significant differences (*p* < 0.05) compared to controls are denoted by *. Legend: lowercase letters a, and b indicate significant differences (*p* < 0.05) between 7, 14, and 21 days of exposure, respectively.

**Figure 8 biomimetics-09-00303-f008:**
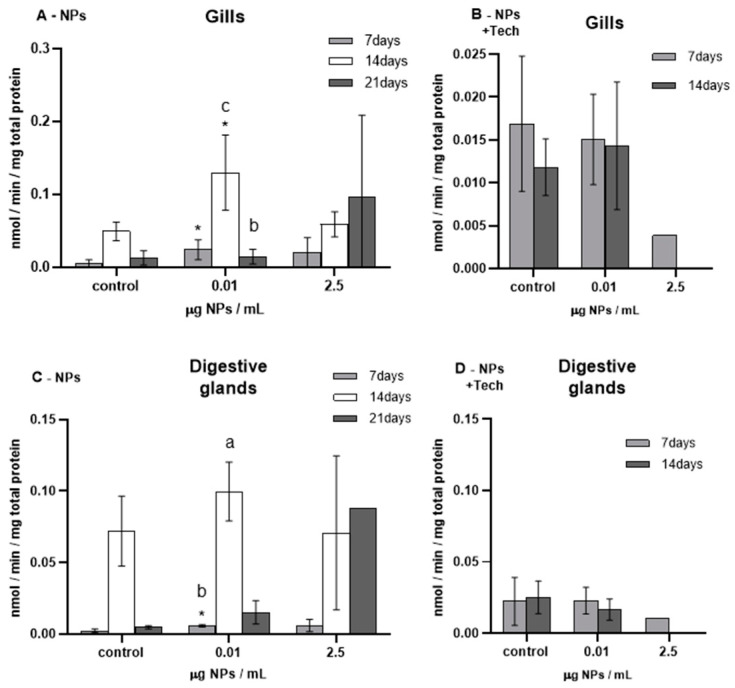
Total ubiquitin (mean ± sd) in *M. galloprovincialis* exposed to single NPs and NPs using the aggregation method at different concentrations measured in gills (**A**,**B**) and digestive glands (**C**,**D**) after 7, 14, and 21 days. Significant differences (*p* < 0.05) compared to controls are denoted by *. Legend: lowercase letters a, b, and c indicate significant differences (*p* < 0.05) between 7, 14, and 21 days of exposure times, respectively.

**Figure 9 biomimetics-09-00303-f009:**
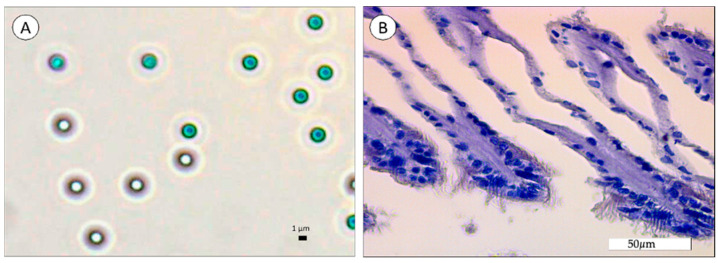
Microscopic observation of PS MPs (**A**) and histological section (**B**) of the gills of *M. galloprovincialis* after 21 days of exposure to 1 µm PS MPs.

**Figure 10 biomimetics-09-00303-f010:**
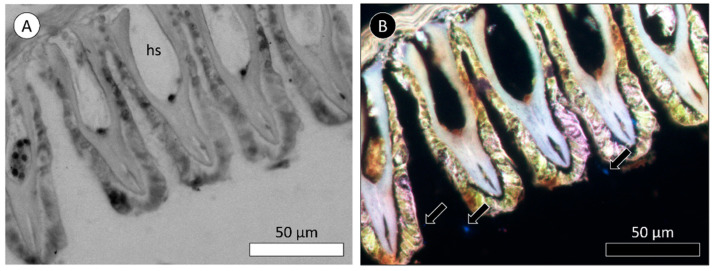
Bright-field (**A**) and fluorescence (**B**) photomicrographs of a section of the gills of *M. galloprovincialis* after 7 days of exposure to 50 nm PS-NH_2_ NPs using the developed technology or method. Legend: hs indicates the hemolymphatic sinus; arrows indicate the locations of NP aggregates.

**Table 1 biomimetics-09-00303-t001:** DLS measurements of NPs in FMQ in different concentrations of FIL and chitosan. Data are presented as (mean ± sd).

NPs (mg/mL)	FIL (mg/mL)	Chitosan (µg/mL)	Z-Average (nm)	PDI	ζ-Pot (mV)	E. M. (cm^2^/Vs) ^1^
0.0200	-	-	276.44 ± 36.78	0.51 ± 0.17	0.10 ± 4.26	0.00000073 ± 0.000033
0.0182	132.15	-	1165.48 ± 85.76	0.69 ± 0.11	−1.56 ± 2.70	−0.0000123 ± 0.000021
0.0179	129.79	0.005	991.06 ± 138.43	0.57 ± 0.13	−0.81 ± 2.79	−0.0000064 ± 0.000022
0.0170	123.71	0.019	1595.27 ± 672.54	0.62 ± 0.15	−0.42 ± 1.82	−0.0000033 ± 0.000014
0.0167	121.13	0.025	1142.76 ± 169.72	0.59 ± 0.11	−1.50 ± 2.04	−0.0000117 ± 0.000016

^1^ E. M.: electrophoretic mobility.

**Table 2 biomimetics-09-00303-t002:** DLS measurements of NPs in FSSW in different concentrations of FIL and chitosan. Data are presented as (mean ± sd).

NPs (mg/mL)	FIL (mg/mL)	Chitosan (µg/mL)	Z-Average (nm)	PDI	ζ-Pot (mV)	E. M. (cm^2^/Vs) ^1^
0.02	-	-	63.66 ± 4.01	0.18 ± 0.11	27.42 ± 10.42	0.000213 ± 0.000081
0.019	69.22	-	3942.48 ± 241.95	0.68 ± 0.28	−11.96 ± 3.03	−0.000093 ± 0.000023
0.0182	132.15	-	4552.53 ± 613.68	0.81 ± 0.11	−13.64 ± 1.22	−0.000106 ± 0.000010
0.0196	-	0.006	73.29 ± 3.99	0.30 ± 0.08	−13.76 ± 1.09	−0.000107 ± 0.000009
0.0186	-	0.021	73.19 ± 3.44	0.30 ± 0.08	38.39 ± 6.80	0.000298 ± 0.000053
0.0182	-	0.027	76.51 ± 8.70	0.32 ± 0.14	17.81 ± 2.16	0.000138 ± 0.000017
0.0179	129.79	0.005	383.80 ± 97.87	0.45 ± 0.08	−14.99 ± 1.51	−0.000116 ± 0.000012
0.0170	123.71	0.019	8322.68 ± 2123.15	0.88 ± 0.06	−7.27 ± 1.34	−0.000057 ± 0.000010
0.0167	121.13	0.025	5658.27 ± 1336	0.79 ± 0.18	−18.23 ± 9.82	−0.000141 ± 0.000076

^1^ E. M.: electrophoretic mobility.

## Data Availability

Data supporting this study are included within the article and Appendix A.

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
