# Peer review of "Development of a New Aggregation Method to Remove Nanoplastics from the Ocean: Proof of Concept Using Mussel Exposure Tests"

_biomimetics, 2024, doi:10.3390/biomimetics9050303_

Round 1
Reviewer 1 Report
Comments and Suggestions for Authors
biomimetics-2998673
Title: Development of a new aggregation method to remove nanoplastics from the ocean: proof of concept using mussel exposure tests
The manuscript “Development of a new aggregation method to remove nanoplastics from the ocean: proof of concept using mussel exposure tests” aims to achieve the aggregation of micro and nanoplastics (MNPs) to ease their removal from the marine environment. The environmental significance of the paper is good and the content is relatively comprehensive, but the writing, data analysis and figures need to be improved. Therefore, the manuscript is subject to considerable revision before it is suitable for publication in this journal. Here are some suggestions for the authors.
1. The background on PS microplastics and the sources and production processes of microplastics is too verbose and illogical.
2. Why is there a special focus on microplastics in the Marine environment, ecotoxicology, and current remediation techniques for micro(nano)plastics? Instead of writing directly in the introduction.
3. There are many kinds of surfactants, what is the basis for choosing ionic liquids as surfactants?
4. A method of polymerization with surfactant (FIL) and polyelectrolyte (chitosan) was proposed to simulate the behavior of NOM adsorption on the surface of MNPs under actual environmental conditions, whether this is representative. Natural organic matter is a complex mixture that is different from surfactants.
5. How is the Mediterranean mussel obtained, and the specific information about its individual size and age needs to be detailed.
6. The main information of the material of the 0.20µm filter needs to be written clearly. The Dh is defined as the diameter of a hypothetical sphere that diffuses at the same rate as the particle. Why is the serial number of the formula (5)?
7. What is the basis of the polymerization method, and the characterization methods of Dynamic Light Scattering (DLS), zeta-potential and SEM are not enough.
8. Figures 2-5 are very vague and confused, it is recommended to reproduce the drawing and put these figures together.
9. 3.1.1. Microplastics and Nanoplastics Characterization. There are too many small paragraphs, and the characterization of microplastics and nanoplastics can be put together, and at most two paragraphs can be discussed.
10. 3.2.1. Mortality and 3.3. Histological observations. The content here is not thin enough to be the content of a subtitle, it is recommended to add the result analysis.
11. The contents of the discussion are long but not logical, and many contents are repeated. It is suggested to refer to the discussions of other papers for revision.
12. The conclusion is an important summary of the content of this study, which summarizes the important results, conclusions and research significance of this paper, and also points out the shortcomings of this paper. It is suggested to resummarize, without breaking it into four paragraphs.
Comments on the Quality of English Language
Moderate editing of English language required
Author Response
The authors greatly appreciate the reviewers' comments on improving the manuscript.
- Answer: This section has been summarized and made more logical at the reviewer's suggestion.
- Answer: Following the reviewer's suggestion, this section has been included directly in the introduction.
- Answer: the ionic liquids have been selected due to the charge provided by the sulfonate group, which provides us with the charge to reach the point of zero charge and, therefore, destabilize the colloidal dispersion of Nanoplastics and, thus, promote its aggregation.
- Answer: Natural organic matter is formed by a complex mixture, but its major components are humic and fulvic acids, which exhibit behavior similar to surfactants. Through self-assembly, they can form micelles or microemulsions, so this behavior helped us hypothesize that we could replicate it when adsorbed on the surface of nanoplastics. The heteroaggregation method is described in lines 81-134 (original manuscript) and explained and discussed based on results and previous research in lines 749-784 (original manuscript).
- Answer: Mussels were manually collected in Praia da Parede (Cascais, Lisbon, Portugal). The shell length of the mussels was 3.73 ± 0.25 cm. The age of the mussels was not determined since it is difficult to assess; therefore, most studies usually use the size. For example, sexual maturity is normally related to body size rather than age.
- Answer: The type of filters used has been specified. Thank you for pointing out the typographical error; it has been corrected.
- Answer: The basis of the aggregation method is described in lines 81-134 (original manuscript). It is explained that colloidal dispersions of nanoplastics are destabilized under the appropriate conditions. Therefore, their aggregation is promoted and governed by electrostatic and steric interactions, which makes the nanoplastics less bioavailable.
- Answer: the reviewer asks us to unify all the figures, but this may need clarification. Furthermore, some are prepared using MilliQ water, and others are synthetic and filtered seawater, so it makes no sense. The best idea would be to eliminate the figures since they still repeat the data that appears in Tables 1 and 2.
- Answer: Following the reviewer's recommendations, the paragraphs have been unified and reduced.
- Answer: Following the reviewer's recommendations, the highlighted subheadings have been removed, and the results have now been commented on.
- Answer: The authors greatly appreciate the reviewer's comment. The discussion refers to the exposure of mussels to both MPs and NPs, highlighting the importance of the size of the plastics, as well as the efficiency of our aggregation method depending on the size and surface charge they present, which plays a role essential when it comes to making the aggregation of NPs more effective, being less effective in the case of micrometric plastics. What makes the aggregation method manage to increase the size of the heteroaggregates of NPs to micrometric size led us to conclude that the NPs alone had a more significant effect on the biomarkers analyzed when they were dispersed than when we managed to form the aggregates. To shorten this part of the discussion suggested by the reviewer, we could move the MPs part to the supplementary material.
- Answer: Following the reviewer's recommendations, the text of the conclusions has been unified into a single paragraph.
Reviewer 2 Report
Comments and Suggestions for Authors
This paper describes a new aggregation method to remove nanoplastics from the ocean. This is a relevant scientific paper. There are some problems with the analysis of the results because the errors of the experimental measurements are relatively high, which not allow the relative comparison of the averages because they are statistically similar.
- Lines 247-249. A citation must be given about the origin of the FIL.
- Lines 450-451; Lines 479-482; etc. The classification as "highest" value must take into consideration the corresponding error. There are many cases where the error is enormous and this classification cannot be done. It must be corrected.
Author Response
The authors greatly appreciate the reviewers' comments on improving the manuscript.
1- Lines 247-249. A citation must be given about the origin of the FIL.
Answer: Following the reviewer's suggestion, the proper reference was added.
2- Lines 450-451; Lines 479-482; etc. The classification as "highest" value must take into consideration the corresponding error. There are many cases where the error is enormous, and this classification cannot be done. It must be corrected.
Answer: As results are expressed in mean ± s.d., not only are the mean values taken into account, but the statistical analyses served us to establish significant differences between control and exposure concentrations to NPs, so the sentence takes all of these parameters into account.
Reviewer 3 Report
Comments and Suggestions for Authors
The micro-nano plastics are actual issue and problem in the marine environment and seafood. The manuscript (MS) addressed a new approach and aspects to detect and remove micro-nano plastics using a strong suspended filterer (M. galloprovincialis) basing the procedures on observation of effect of the plastics on its physiology, biochemical and histological features with a bioassay study.
The MS suggested a new guide for the detection and removal of the micro-nanoplastics via the filterer, which is interested to wide readers and audience. Up to date, there were many studies on quantitative and quantitative distribution of the plastics in the aquatic environment. Nobody touched this matter to remove the matter. Recently, experts on plastics must be involved to solution of the problem. This study fill the gap purposed for the matter.
The MS was subjected to a technological approach to detect MNP in size and abundance and a variety of biochemical, anatomical (gill) and histological measurements. Such context was achieved for the first time for such purposed study. The further study could be applied to this approach conducted in situ rather than ex situ studies according to methodology of the present study.
The methodology has not been considered currently with present form of the MS which would be diversified of the filtering organisms in the marine environment.
The empirical study with bioassay supported all significance of the all variable measurements to conclude aim of the MS. The plastics are popular problem of nowadays in the marine environment.
The references used in the MS is sufficient enough.
see the following entire comments:
The MS was well described from Abstract through results to Discussion. The MS was really long particularly Introduction, followed by Material and methods. Some of Material and Methods could be moved to existed Supplementary file which was also long or vice versa, e.g. statistical analyses could be moved to Material and Methods.
The Results ar5e full of number of measurements with Table and Figures as well. I suggested the numbers could be tabulated in Table or Figure enough, followed by a brief results instead of number, significantly higher or lower etc.
In the text there are many abbreviations, an abbreviation list at the beginning could better.
At the end of results or each section of results a summary result would better for better understanding. The readers could be missed among the many numbers of measurements.
For each table and Figure results could be subjected to statistical test of the significance for difference for each factors or on X axis on the figure.
Conclusion must emphasize importance and novelty of the MS.
Minor
Labels and legendary of the Figures were not clear.
English of the text language could be edited and reviewed using a creative way.
The text strength could be shortened by 1/3, remaining 2/3.
Indeed, the MS was scientifically accepted for a publication, but the MS could be accepted after major revision in order for the MS to be improved and better understanding.
Comments on the Quality of English LanguageEnglish of the text language could be edited and reviewed using a creative way.
The text strength could be shortened by 1/3, remaining 2/3.
Author Response
The authors greatly appreciate the reviewers' comments on improving the manuscript.
Answer:
Following the reviewers' suggestions, the text has been shortened, and part of the manuscript has been moved to supplementary material, thus reducing its size.
In the text there are many abbreviations, an abbreviation list at the beginning could better.
Answer: A list of abbreviations has been added to the beginning of the manuscript.
At the end of results or each section of results a summary result would better for better understanding. The readers could be missed among the many numbers of measurements.
For each table and Figure results could be subjected to statistical test of the significance for difference for each factors or on X axis on the figure.
Conclusion must emphasize importance and novelty of the MS.
Answer: Following the reviewers' suggestions, the results have been presented in a more summarized form, and the manuscript's importance has been underlined.
Minor
Labels and legendary of the Figures were not clear.
Answer: Following the reviewers' suggestions, the graphs' labels and legends are improved.
English of the text language could be edited and reviewed using a creative way.
Answer: Following the reviewers' suggestions, the text has been revised and improved.
The text strength could be shortened by 1/3, remaining 2/3.
Indeed, the MS was scientifically accepted for a publication, but the MS could be accepted after major revision in order for the MS to be improved and better understanding.
Answer: Following the reviewers' suggestions, we have summarized and improved the manuscript's text.
Round 2
Reviewer 1 Report
Comments and Suggestions for Authors
accept
Comments on the Quality of English LanguageMinor editing of English language required
Reviewer 3 Report
Comments and Suggestions for Authors
The manuscript (MS) was improved as compared to the previous version. However, the English of the text was not overviewed. Therefore, I suggest that the Editorial of the journal could decide checking the English if the MS is accepted. The text was insignificantly shortened. Few figures were removed from the text only.
The MS could be accepted with present version.